# Joint Pixel-Token Compression for Efficient Video-Language Models

## Abstract

Recent advances in Video Large Language Models (VLLMs) have significantly enhanced multimodal comprehension capabilities. However, the substantial computational demands of these models remain a major bottleneck, primarily due to the extensive number of video frames and the correspondingly large volume of visual tokens produced by video encoders. Conventional approaches often rely on uniform frame sampling, which captures abundant but highly redundant visual information: at the pixel level, adjacent frames frequently contain overlapping content, failing to adequately represent temporal dynamics; at the token level, numerous visual tokens encode repetitive patterns, resulting in unnecessary computational overhead. To address these challenges, we propose a novel joint Pixel–Token (P-T) compression strategy designed to reduce computational costs while preserving semantic richness. At the pixel level, our method assesses inter-frame similarity enabling the selection of semantically informative frames that better capture temporal variations. At the token level, we compress the visual representation by computing similarity between corresponding tokens across frames, thereby eliminating redundant tokens without compromising critical information. The proposed framework functions as a plug-and-play module, ensuring easy integration into various baseline models. Extensive experiments under both training-free and fine-tuning scenarios validate the effectiveness of our approach. Remarkably, even after discarding more than 50% of visual tokens, our method achieves a 0.9% performance gain on the MVBench benchmark.

## 1 Introduction

Video Large Language Models (VLLMs) have witnessed remarkable progress in comprehending complex video content, achieving state-of-the-art performance across a variety of tasks including question answering, video captioning, and temporal grounding Li et al. (2023); Chen et al. (2024b); Share (2024); Guo et al. (2024). Despite these advances, processing long videos remains computationally expensive: a large number of frames must be fed into the language model along with the textual prompt, and each frame is encoded by the visual encoder into thousands of tokens. This results in substantial computational overhead and introduces considerable redundancy in visual information. Developing strategies to effectively reduce redundant visual tokens while preserving model performance thus constitutes a critical and challenging problem in advancing VLLM efficiency and scalability.

Visual redundancy in videos arises from multiple sources. On the one hand, videos are traditionally processed using uniform frame sampling, which selects frames at fixed intervals regardless of their content. While simple, this approach often introduces a large amount of redundant visual information, especially when consecutive frames are highly similar in appearance, leading to unnecessary data duplication. On the other hand, after passing through a visual encoder, each frame is transformed into a very large number of visual tokens. Many of these tokens are semantically similar or redundant, resulting in an excessive amount of information that must be processed. This not only increases the computational cost but also imposes a significant memory overhead on subsequent video understanding models. Figure 1(a) illustrates how previous methods handle video input, highlighting the inefficiency caused by both redundant frames and redundant token representations. Consequently, effectively reducing visual redundancy while maintaining the original model's performance

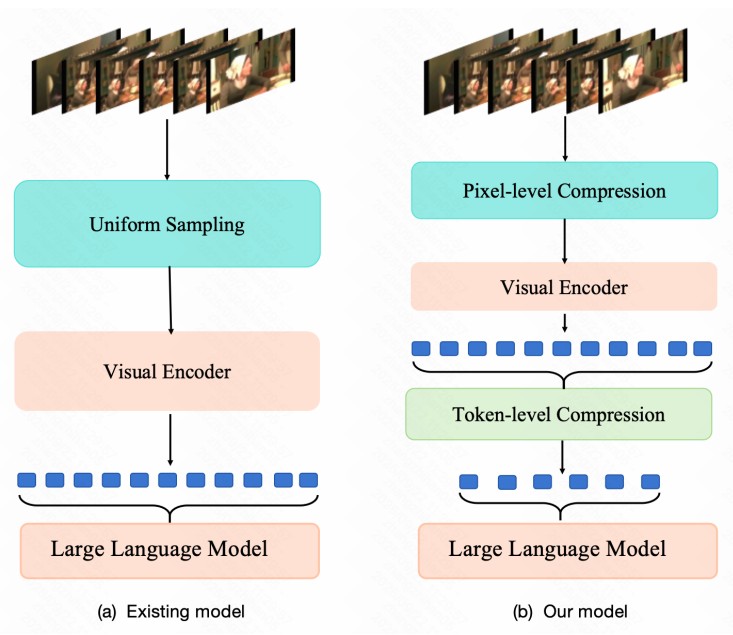

Figure 1: Comparsion between the exsiting model and our model

remains a critical and challenging problem, as existing strategies fail to balance computational efficiency with the preservation of informative content.

To address this issue, we propose a joint pixel-token compression strategy that effectively reduces redundant visual information while preserving essential semantic content. Figure 1(b) shows the method of our model. Specifically, our method operates in two complementary stages: pixel-level compression and token-level compression, forming a unified pipeline that efficiently prunes redundant information at both the frame and token granularity. Firstly, we employ pixel-level compression to select semantically relevant frames from the videos. We measure the similarity between consecutive frames by computing their pixel-wise differences. This process is performed iteratively, and when the difference falls below a predefined threshold, the second frame is regarded as redundant; otherwise, it is retained. Such a strategy effectively eliminates visually redundant frames, thereby reducing computational overhead and improving the efficiency of subsequent video understanding. Secondly, we adopt the token-level compression to further reduce the redundant tokens. We eliminate unnecessary visual information by computing similarity between corresponding tokens across frames, removing tokens with high similarity while retaining those with low similarity. Also, we also apply a dynamic pruning rate to better select tokens according to their semantic relevance, ensuring a more appropriate set of tokens is retained. We can effectively reduce the computation burden while still maintain the model performance through this joint pixel-token compression method. We conduct extensive experiments under both training-free and training settings, achieving comparable or superior performance across multiple benchmarks and diverse baselines.

Our contribution can be summarized as follows:

- We propose a plus-and play joint pixel-token compression strategy to reduce the computation cost at different dimensions, achieving effective video understanding while significantly reducing the computational burden.

- We introduce the pixel-level compression strategy to select more semantically informative frames. Also, we employ token-level compression approach to eliminate unnecessary visual information by computing the similarity at corresponding positions between frames.

- We extensively evaluate our approach under both training-free and training conditions, achieving competitive or superior results across multiple benchmarks and different baseline models.

## 2 RELATED WORK

### 2.1 MULTI-MODAL LARGE LANGUAGE MODELS

Recently, Multimodal Large Language Models (MLLMs) Zhang et al. (2025a); Bai et al. (2025) have made remarkable progress in video understanding. An increasing number of studies have explored integrating LLMs to leverage their capabilities for video tasks such as visual question answering, captioning, grounding, and reasoning Zhang et al. (2023); Lin et al. (2023); Zhang et al. (2023); Chen et al. (2024b); Cheng et al. (2024). For example, VideoChat Li et al. (2023) bridges video foundation models and LLMs through a learnable interface, demonstrating strong potential in spatiotemporal reasoning and event localization. LLaVA-OneVision Li et al. (2024a) and LLaVA-Video Zhang et al. (2024b) extend this paradigm to support single-image, multi-image, and video scenarios by exploiting diverse data sources. TRACE Guo et al. (2024) introduces a causal event modeling framework to precisely localize timestamps for temporal grounding. SpaceVLLM Wang et al. (2025) is designed to investigate and enhance the spatio-temporal video grounding capability of multimodal large language models. Video-R1 Feng et al. (2025) proposes the T-GRPO algorithm to enhance video reasoning abilities. Despite these advances, passing thousands of video tokens into LLMs imposes significant computational overhead as the number of frames increases. Thus, an open challenge is how to effectively reduce redundant information while preserving the original performance.

### 2.2 VISUAL INFORMATION COMPRESSION.

Several studies Bolya et al. (2022); Xing et al. (2024); Yang et al. (2025) have explored methods for compressing visual information, which can be broadly categorized into training-free and training-based approaches. Among training-free methods, FastV Chen et al. (2024a) enhances attention efficiency in MLLMs by pruning redundant image tokens based on attention scores. LLaVA-PruMerge Shang et al. (2024) performs adaptive token reduction by retaining key visual tokens identified through attention scores from the CLIP visual encoder. DyCoke Tao et al. (2025) is a training-free method that accelerates VLLMs through dynamic merging of redundant temporal tokens and selective pruning of unnecessary spatial tokens. In the training-based category, HICom Liu et al. (2025b) employs a hybrid-level conditional token compression strategy that leverages instructions to preserve user-relevant visual information. LongVU Shen et al. (2024) is a spatiotemporal adaptive compression mechanism that reduces the number of video tokens while preserving visual details of long videos. Video-XL-Pro Liu et al. (2025a) utilizes reconstructive token compression to generate compact yet informative video tokens while adaptively masking redundant ones. Additionally, Q-Frame Zhang et al. (2025b) adaptively selects and scales video frames according to content and query relevance using a text-image matching network. Despite these advances, no existing work simultaneously compresses visual information at both the pixel and token levels.

## 3 METHOD

In this section, we first overview our architecture and then detailedly illustrate our plus-and play joint pixel-token compression approach.

### 3.1 OVERVIEW

The overall architecture of our method is illustrated in Figure 2. We begin by sampling a substantial number of frames from the video to ensure sufficient visual coverage. Next, we adopt a pixel-level compression strategy to filter out redundant frames and retain those that are more semantically informative. The selected frames are then processed by a visual encoder to obtain video tokens. To further eliminate redundancy, we apply a token-level compression strategy, yielding a more compact yet informative token representation. Finally, the refined video tokens are fed into a large language model to perform downstream video understanding tasks.

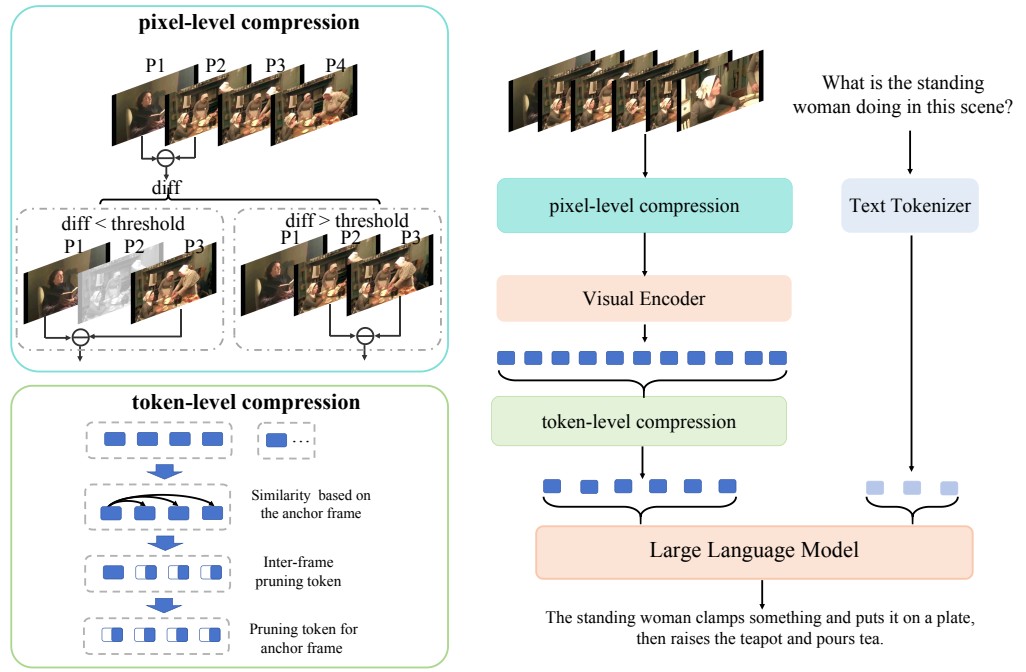

Figure 2: The overall architecture of joint pixel-token compression method. Videos first pass through the pixel-level compression to select the semantic relevant frames. Token-level compression aims to further eliminate redundant video tokens by measuring the similarities across the frames.

## 3.2 PIXEL-LEVEL COMPRESSION

We begin by sampling a sequence of video frames, denoted as $F = \{f_0, f_1, \ldots, f_{n-1}\}$. In the pixel-level compression stage, the objective is to filter out visually redundant frames while preserving those that carry meaningful semantic information. To this end, we evaluate the similarity between consecutive frames by computing the $L_1$ distance at the pixel level. Specifically, the difference between two adjacent frames is quantified through their pixel-wise discrepancies, providing a simple yet effective measure of visual similarity. This computation is performed iteratively along the temporal axis. Starting from an initial retained frame, each subsequent frame is compared against the most recently preserved frame. If the difference falls below a predefined threshold $\tau$, the frame is deemed redundant and discarded; otherwise, it is retained for downstream processing. The detailed algorithm is illustrated in Algorithm 1.

By applying this pruning mechanism, we substantially reduce the number of redundant frames while ensuring that the retained frames remain semantically informative. This selective preservation not only alleviates the computational burden in later processing stages but also enhances the effectiveness of video understanding by focusing on frames that capture meaningful changes in the visual scene.

## 3.3 TOKEN-LEVEL COMPRESSION

In the token-level compression, we further eliminate redundancy in the feature representations obtained from a visual encoder. Each retained frame is encoded into a set of visual tokens, many of which may carry semantically overlapping information. This compression is designed to effectively eliminate redundant video tokens after the visual encoder processes the video frames, thereby reducing computational overhead while preserving essential information. Redundant visual information is removed by measuring the cosine similarity between tokens at corresponding positions across frames, where highly similar tokens are discarded and less similar ones are preserved.

---

**Algorithm 1:** Iterative Pixel-level Frame Pruning

---

**Input:** Frames $\{f_0, f_1, \ldots, f_{n-1}\}$, threshold $\tau$
**Output:** Retained frame indices $\{t_0, t_1, \ldots\}$
$t \leftarrow 0$; **while** $t < N - 1$ **do**
    |    $j \leftarrow t + 1$;     found $\leftarrow$ False;
    |    **while** $j \leq N - 1$ **do**
    |    |    **if** $d(f_t, f_j) \geq \tau$ **then**
    |    |    |    retain.append($j$);
    |    |    |    $t \leftarrow j$;
    |    |    |    found $\leftarrow$ True;
    |    |    |    **break**;
    |    |    **else**
    |    |    |    $j \leftarrow j + 1$;
    |    **if** *not found* **then**
    |    |    **break**
**return** retain

---

In detail, we first partition the frame-level tokens into groups using a sliding window and compute the similarity between tokens at corresponding positions across adjacent groups. Within each group, the first frame is designated as the anchor frame, and the cosine similarity between the anchor frame and the other frames in the group is calculated. The token-wise similarity score can be formulated as:

$$S(i,j) = \cos(\theta) = \frac{h_i \cdot h_j}{\|h_i\|\|h_j\|} \tag{1}$$

where $h$ represents the visual embedding. $S$ refers to the cosine similarities at corresponding positions between frames.

Within each sampling window, we prune highly similar tokens. all tokens from the first frame of the window are preserved in their entirety, as this frame serves as the anchor and provides a complete semantic reference. For the subsequent frames within the same window, tokens are selectively pruned according to their similarity to the anchor tokens. This design ensures that the most representative frame maintains full spatial information, while redundant information from neighboring frames is effectively compressed. Furthermore, the pruning ratio is not fixed but dynamically adjusted to better adapt to variations across different video segments. To achieve this, we introduce a similarity-based thresholding mechanism. For each token, we compute its similarity score with respect to the corresponding anchor token. If the similarity score is below a predefined threshold $\tau$, the token is considered informative and is therefore preserved; otherwise, it is pruned. In addition, the pruning ratio is constrained within a controlled range $[\rho_{\min}, \rho_{\max}]$, ensuring that the method maintains an appropriate balance between efficiency and information retention.

$$\mathcal{T}_{\text{retain}} = \{t_i \mid \text{sim}(t_i, t_{\text{anchor}}) < \tau\}, \quad \rho \in [\rho_{\min}, \rho_{\max}], \tag{2}$$

where $\text{sim}(\cdot)$ denotes the cosine similarity function, $t_i$ is the token from the current frame, and $t_{\text{anchor}}$ is the corresponding token from the anchor frame. The dynamic pruning ratio $\rho$ guarantees that the retained token set $\mathcal{T}_{\text{retain}}$ remains both semantically informative and computationally efficient. Through this pruning intra-frame strategy, we can effectively remove the redundant video tokens through the connections across the frames.

For the anchor frame, we aim to identify and prune tokens that exhibit excessive redundancy with respect to other tokens in the same frame. To achieve this, we first compute an aggregated similarity score for each token, defined as the sum of its pairwise similarities with all the other tokens in the frame.

$$s_i = \sum_{j=1, j \neq i}^{N} S(t_i, t_j), \tag{3}$$

where $t_i$ denotes the $i$-th token, $N$ is the total number of tokens in the anchor frame, and $S(\cdot, \cdot)$ represents the cosine similarity function. A larger $s_i$ indicates that token $t_i$ is more redundant, as it exhibits higher similarity with other tokens. To prune redundant tokens, we sort all tokens by their redundancy scores and preserve only the least redundant ones. Specifically, the retained token set $\tilde{T}$ is defined as

$$\tilde{T} = \text{Top}_{(1-r)N}^{\text{lowest}}\{(t_i, s_i)\}_{i=1}^N \tag{4}$$

where $r$ is the pruning ratio. The dynamic pruning process ensures that the preserved set $\tilde{T}$ maintains semantic diversity while reducing computational overhead. The token-level compression ensures that the model focuses on the most informative visual tokens, thereby accelerating inference without sacrificing accuracy. The proposed joint pixel-token compression method effectively reduces both temporal and spatial redundancies while maintains the model's performance.

## 4 EXPERIMENTS

### 4.1 EXPERIMENT SETTINGS

#### 4.1.1 IMPLEMENTATION DETAILS

To evaluate the effectiveness of our proposed joint pixel-token compression method, we integrate it into two widely-used multimodal baselines, LLaVA-Video Zhang et al. (2024b) and Qwen2.5-VL Bai et al. (2025). In our experiments, we sample 64 frames from each video as the input for both baselines. For the pixel-level compression, we set the similarity threshold $\tau$ to 0.1. At the token level, we adopt a compression threshold of 0.5 and dynamically constrain the compression ratio between $\rho_{\min} = 0.5$ and $\rho_{\max} = 0.7$. For the training-based variant of our method, we conduct a single epoch of fine-tuning on the model using 8 NVIDIA A800 GPUs.

#### 4.1.2 DATASETS AND BENCHMARKS

To further explore the potential of compression for improving model performance, we collect video data from several sources, including a sampled subset of LLaVA-Video 178K Zhang et al. (2024b), STAR Wu et al. (2024), Charades Gao et al. (2017), PerceptionTest Patraucean et al. (2023), and Clevrer Yi et al. (2020), resulting in a total of 120K videos. To comprehensively evaluate the effectiveness of our approach, we employ the LMMs-Eval Zhang et al. (2024a) for evaluation on multiple widely-used video understanding datasets, including MVBench Li et al. (2024b), VideoMME Fu et al. (2024), and NextQA Xiao et al. (2021) respectively.

### 4.2 MAIN RESULTS

#### 4.2.1 TRAINING-FREE SETTINGS

First, we conduct experiments under training-free settings to evaluate the effectiveness of our proposed approach. First, we compare different training-free methods. We utilize the LLaVA-OV 7B model for fair comparison. As shown in Table 1, under a comparable compression ratio, our pixel-token (75%) method consistently outperforms Dycoke (77%) across all benchmarks. This demonstrates the effectiveness of our joint pixel-token compression in reducing redundancy while preserving temporal and semantic information. Then, we apply two widely used baseline models, LLaVA-Video and Qwen2.5-VL. We can observe that dynamically adjusting the pruning ratio of visual information leads to consistently superior performance compared with uniform frame sampling. For instance, on the MVBench benchmark, our method achieves a 0.7% improvement over uniform sampling, even though more than 50% of the visual tokens are pruned. This demonstrates that our approach is able to adaptively preserve the most semantically informative frames while discarding redundant information. Furthermore, both pixel-level and token-level pruning strategies achieve comparable or even better results across multiple benchmarks under the two baseline models. For example, on the VideoMME benchmark, the proposed method under Qwen2.5-VL baseline yields a 0.9% gain compared with uniform sampling, despite removing half of the frames. This suggests that the selected frames already capture the essential semantic content of the video, thereby validating the robustness of our approach.

Table 1: Results comparison between different baselines under the training-free settings.

| LLaVA-OV 7B | | | |
|---|---|---|---|
| Method | VideoMME w/o subtitle | VideoMME w/ subtitle | NextQA |
| FastV(65%) Chen et al. (2024a) | 57.3 | 60.5 | 78.2 |
| PruMerge(45%) Shang et al. (2024) | 52.9 | 57.0 | 76.0 |
| Dycoke(77%) Tao et al. (2025) | 58.8 | 61.0 | 79.1 |
| pixel-token(ours)(75%) | **59.5** | **63.0** | **79.3** |

| LLaVA-Video | | | | |
|---|---|---|---|---|
| Method | MVBench | VideoMME w/o subtitle | VideoMME w/ subtitle | NextQA |
| Uniform 64 frames | 61.0 | 59.1 | 69.6 | 81.5 |
| Pixel (50%) | 61.6 | **61.0** | 69.6 | **82.5** |
| Token (50%) | 61.4 | 60.0 | 69.1 | 82.1 |
| $\tau$=0.5 (50%–70%) | **61.7** | 59.7 | **69.8** | 81.8 |

| Qwen2.5-VL | | | | |
|---|---|---|---|---|
| Method | MVBench | VideoMME w/o subtitle | VideoMME w/ subtitle | NextQA |
| Uniform 64 frames | 67.2 | 63.3 | 67.9 | 75.1 |
| Pixel (50%) | 66.8 | 63.5 | 69.0 | 75.6 |
| Token (50%) | 67.6 | 63.7 | 69.6 | 76.3 |
| $\tau$=0.5 (50%–70%) | **68.1** | **64.5** | **69.7** | **76.9** |

### 4.2.2 TRAINING-BASED SETTINGS

Moreover, we conduct extensive experiments under training-based settings to further validate the effectiveness and robustness of our model. As presented in Table 2, the results show that our approach consistently achieves comparable or even superior performance, despite discarding up to 50% of the visual information. For example, on the MVBench benchmark, our pixel-level compression strategy under the LLaVA-Video baseline delivers a 1.7% improvement compared to the uniform sampling strategy. This clearly demonstrates that our method is capable of selecting more semantically meaningful frames, which in turn enhances video understanding. Similarly, our token-level compression strategy achieves a 1.4% gain on the Qwen2.5-VL model on the VideoMME benchmark, even when pruning half of the visual tokens. This result indicates that a significant portion of visual tokens is redundant and may even hinder model performance if left unfiltered. Furthermore, our joint pixel-token compression strategy maintains competitive results even under more aggressive pruning. Notably, when eliminating 75% of the visual information, our model still achieves a 1.1% improvement on the NextQA benchmark under Qwen2.5-VL, further confirming its ability to retain the most relevant semantic cues while discarding irrelevant information.

In summary, these training-based experiments strongly support the conclusion that our proposed compression strategies—whether applied at the pixel level, token level, or jointly—can significantly reduce the amount of visual information without sacrificing, and in some cases even enhancing, performance. This finding highlights the importance of identifying and removing redundant inputs in large-scale video understanding tasks.

### 4.3 ABLATION STUDIES

In this section, we conduct different ablation studies to validate the effectiveness of our model. In this part, we experiment with the training-free settings.

### 4.3.1 ARCHITECTURE ANALYSIS

First, we conduct ablation studies on the model architecture to investigate the effectiveness of our proposed compression strategies. In these experiments, we apply the same pruning ratio across three settings—pixel-level compression, token-level compression, and joint pixel-token compression—to ensure a fair comparison of their impact. The main objective is to examine whether combining both pixel- and token-level pruning can better preserve semantically relevant visual information that is essential for video understanding. As shown in Table 3, the results on the MVBench benchmark under both the LLaVA-Video and Qwen2.5-VL baselines consistently demonstrate the superiority of our joint compression strategy. Specifically, our method achieves 0.4% and 1.1% improvements

Table 2: Results comparison between LLaVA-Video and Qwen2.5-VL on multiple benchmarks under the training-based settings.

| LLaVA-Video | | | | |
|---|---|---|---|---|
| **Method** | **MVBench** | **VideoMME w/o subtitle** | **VideoMME w/ subtitle** | **NextQA** |
| Uniform 64 frames | 62.8 | 59.1 | 69.6 | 81.5 |
| Pixel (50%) | **64.5** | **61.0** | 69.6 | **82.5** |
| Token (50%) | 64.0 | 60.0 | 69.1 | 82.1 |
| Pixel-Token (75%) | 63.9 | 58.7 | **69.8** | 81.8 |
| **Qwen2.5-VL** | | | | |
| **Method** | **MVBench** | **VideoMME w/o subtitle** | **VideoMME w/ subtitle** | **NextQA** |
| Uniform 64 frames | 68.6 | 60.1 | 63.9 | 79.2 |
| Pixel (50%) | 68.6 | 60.7 | 64.9 | **81.2** |
| Token (50%) | **69.0** | **61.5** | **65.1** | 81.0 |
| Pixel-Token (75%) | 68.3 | 60.5 | 64.4 | 80.3 |

over pixel-level compression, and 0.5% and 0.6% improvements over token-level compression on the two baselines, respectively. These consistent gains highlight that pixel-level and token-level information are complementary: pruning them separately tends to discard distinct aspects of semantic content, whereas their joint integration more effectively captures the essential frames and tokens that contribute to downstream reasoning. Moreover, the results show that even when up to 75% of visual information is removed, the joint strategy is still able to maintain comparable performance to the original model. This observation indicates that a substantial portion of the raw visual inputs are redundant and may not directly contribute to semantic understanding. By contrast, our joint pruning strategy adaptively filters out less informative visual content while retaining those frames and tokens that are most semantically critical. In summary, these findings not only validate the effectiveness of our design but also provide new insights into how redundancy exists in both pixel- and token-level representations. The fact that such aggressive pruning can still preserve performance suggests that video-language models do not necessarily require dense visual signals, but rather benefit more from carefully selected, semantically meaningful ones. This insight has practical implications for deploying video-language models in resource-constrained environments, where computational efficiency and memory footprint are as crucial as task accuracy.

Table 3: Performance under different compression strategies (MVBench).

| **Compression strategy** | **LLaVA-Video** | **Qwen2.5-VL** |
|---|---|---|
| Uniform 64 frames | 61.0 | 67.2 |
| Pixel (75%) | 61.0 | 66.4 |
| Token (75%) | 60.8 | 66.9 |
| Pixel-Token (75%) | **61.3** | **67.5** |

### 4.3.2 COMPRESSION RATIO

In this section, we conduct ablation studies to investigate the effect of different compression ratios, as summarized in Table 4. The results indicate that our method remains robust across a wide range of compression levels. Specifically, when the pixel-level compression ratio reaches 50%, the model not only maintains its performance but even surpasses the original baseline, achieving a 0.6% improvement under the LLaVA-Video compared with uniform frame sampling. When the token-level compression ratio reaches 50%, the model still outperforms the uniform frame sampling baseline, yielding a 0.4% improvement. As the ratio increases to 75%, the performance becomes largely comparable to that of the original model, suggesting that the framework can tolerate a relatively high level of compression without significant degradation. However, when the compression ratio is pushed to 90%, the model exhibits a clear decline, highlighting the detrimental effect of excessively pruning visual tokens. This degradation can be attributed to the removal of a substantial amount of semantically informative content, which limits the model's ability to fully capture temporal and

contextual dependencies. Overall, these findings demonstrate that moderate compression strikes a favorable balance between efficiency and accuracy, while overly aggressive compression inevitably harms performance. This observation provides practical guidance for setting compression ratios in real-world applications where both computational efficiency and task performance need to be carefully balanced.

Table 4: Performance under different compression strategies (MVBench).

| Compression ratio | LLaVA-Video | Qwen2.5-VL |
|---|---|---|
| Uniform 64 frames | 61.0 | 67.2 |
| pixel (50%) | **61.6** | 66.8 |
| pixel (75%) | 61.0 | 66.4 |
| pixel (90%) | 58.4 | 63.4 |
| Token (50%) | 61.4 | **67.6** |
| Token (75%) | 60.8 | 66.9 |
| Token (90%) | 59.8 | 65.6 |

### 4.3.3 SIMILARITY MEASURE

In this part, to evaluate the impact of different similarity measures in our token-level compression strategy, we compare cosine similarity, L1 distance, and attention dot product. As reported in Table 5, cosine similarity achieves the best overall performance, reaching 61.4% on LLaVA-Video and 67.6% on Qwen2.5-VL. In contrast, L1 distance yields slightly lower results (60.6% and 66.8%), while attention dot product performs the worst (60.2% and 66.2%). These results consistently demonstrate that cosine similarity is the most effective similarity metric for guiding token pruning. The advantage of cosine similarity stems from its ability to focus on the angular relationship between embedding vectors rather than their absolute magnitudes. Since token embeddings may vary in scale across layers and modalities, relying on L1 distance or attention scores can introduce bias toward tokens with larger norms. By normalizing vector length, cosine similarity emphasizes semantic alignment and ensures that pruning decisions are based on content relevance rather than embedding scale. This makes it particularly robust and generalizable in multi-modal token selection. The ablation studies suggest that cosine similarity not only delivers the strongest empirical performance but also provides a principled and stable criterion for token-level compression, thereby improving the robustness of our pruning strategy across different baselines.

Table 5: Performance under different similarity measure (MVBench).

| Similarity Measure | LLaVA-Video | Qwen2.5-VL |
|---|---|---|
| cosine similarity | **61.4** | **67.6** |
| L1 distance | 60.6 | 66.8 |
| Attention dot | 60.2 | 66.2 |

## 5 CONCLUSION

In this paper, we investigate effective approaches for compressing visual information in video-language models while preserving overall performance. To this end, we propose a joint pixel–token compression strategy that substantially reduces computational overhead without sacrificing accuracy. Specifically, our pixel-level compression module computes frame-wise pixel differences to adaptively select semantically informative frames, thereby avoiding redundancy in temporal sampling. Building on this, our token-level compression strategy further prunes redundant visual tokens by measuring cosine similarity across corresponding token positions, ensuring that only the most semantically relevant representations are retained. A key advantage of our design lies in its plug-and-play nature: the proposed compression module can be seamlessly integrated into different baseline models without additional retraining costs. We conduct extensive experiments under both training-free and training-based settings across multiple benchmarks and achieve outstanding performance, validating the effectiveness of our model.

**Ethics Statement**

This research does not involve human subjects, sensitive personal data, or proprietary/confidential information. All datasets used are publicly available and widely adopted in the community. The work is intended purely for scientific research and poses no foreseeable risks of harmful societal impact. The authors have carefully followed the ICLR Code of Ethics throughout the research and paper preparation process.

**Reproducibility Statement**

The work published in ICLR is reproducible. We have made every effort to ensure the reproducibility of our work. Detailed descriptions of the model architecture, training procedure, and hyperparameter settings are provided in the main text.

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

# A APPENDIX

### A.0.1 USE OF LARGE LANGUAGE MODELS (LLMS)

LLMs are used to aid or polish the writing of this paper. The use was limited to language refinement, and the research ideas, experiments, and analyses were conducted entirely by the authors.

