# OpenReview forum: "Joint Pixel-Token Compression for Efficient Video-Language Models"
_ICLR.cc/2026/Conference — Submitted to ICLR 2026_

### Official Review · Reviewer_BDB3 · 2025-10-29

**Soundness:** 2
**Presentation:** 2
**Contribution:** 2
**Rating:** 2
**Confidence:** 3

**Summary:**

This paper aims to address the problem of redundant visual information in large video-language models. To achieve this goal, the authors propose two compression methods: pixel-level compression and visual token-level compression. Both methods employ a similar technique in which a similarity metric is used to evaluate the information redundancy between adjacent frames or tokens. Frames or tokens with high similarity scores are pruned. Experiments conducted under both training-free and training-based settings demonstrate the effectiveness of the proposed methods.

**Strengths:**

1. The proposed method is straightforward and easy to understand.
2. The algorithm is clearly explained and well-documented.
3. The experiments are comprehensive, covering both training-free and training-based settings.

**Weaknesses:**

The main concern lies in the novelty of the proposed approach. The core idea—pruning frames or visual tokens based on similarity scores—has been extensively explored in prior studies on video compression and visual token reduction. While the paper presents a clear and well-executed implementation, it is not immediately evident how the proposed method substantially advances beyond existing techniques in terms of algorithmic innovation or conceptual contribution. A deeper analysis or clearer differentiation from related work would strengthen the paper’s claim of novelty.

**Questions:**

1. What specific types of information are pruned at the pixel level, and what types are removed at the token level? How are these two forms of compression complementary to each other?
2. In Table 5, the authors examine different similarity metrics. Could other information-theoretic metrics, such as mutual information, also be applied in this context?
3. Similarity-based pruning methods have been extensively explored in prior research. What are the key differences between the proposed algorithm and existing approaches in the literature?

---

### Official Review · Reviewer_fGcp · 2025-10-31

**Soundness:** 3
**Presentation:** 2
**Contribution:** 2
**Rating:** 2
**Confidence:** 3

**Summary:**

The paper proposes a joint pixel–token compression framework to reduce the computational cost of Video LLMs. The key idea is to perform pixel-level compression to remove redundant video frames based on inter-frame similarity, followed by token-level compression that prunes redundant visual tokens across frames using cosine similarity. This dual-stage approach targets both temporal and spatial redundancies. The method is plug-and-play and compatible with popular VLLMs like LLaVA-Video and Qwen2.5-VL, requiring minimal or no retraining. Experiments show that the method can discard over 50% of visual tokens while maintaining or even improving accuracy

**Strengths:**

- Plug-and-play design. The framework can be easily integrated into different VLLMs without architecture modification or retraining.
- Strong empirical validation. Experiments on multiple benchmarks show consistent performance improvements.
- Thorough ablation studies. The authors systematically evaluate the effects of compression ratio, similarity measures, and architectural choices, providing insights into robustness and design trade-offs.

**Weaknesses:**

- Limited technical contribution. The proposed pixel-level and token-level token reduction are well studies in previous works [1-6], making this paper limited contribution.
- Dependence on hyperparameters. Compression thresholds (τ, ρmin, ρmax) may need careful tuning for different datasets, but the paper provides limited discussion on sensitivity or generalization.
- Limited theoretical insight. The method is primarily heuristic, lacking a theoretical explanation or analysis of why the joint compression preserves semantic fidelity.


[1] TimeChat-Online: 80% Visual Tokens are Naturally Redundant in Streaming Videos.
[2] Video Compression Commander:Plug-and-Play Inference Acceleration for Video Large Language Models.
[3] HoliTom: Holistic Token Merging for Fast Video Large Language Models.
[4] LLaVA-Scissor: Token Compression with Semantic Connected Components for Video LLMs.
[5] Multi-Granular Spatio-Temporal Token Merging for Training-Free Acceleration of Video LLMs.
[6] AdaTP: Attention-Debiased Token Pruning for Video Large Language Models.

**Questions:**

no.

---

### Official Review · Reviewer_eJST · 2025-11-01

**Soundness:** 2
**Presentation:** 2
**Contribution:** 2
**Rating:** 2
**Confidence:** 4

**Summary:**

To address the substantial computational demands of video understanding with vision-language large models (VLLMs), this paper proposes a joint pixel-token compression strategy. It combines pixel-level compression, achieved by selecting keyframes based on inter-frame differences, with token-level compression, implemented by pruning redundant visual tokens according to their semantic similarities. The approach is evaluated across three VLLMs, demonstrating its effectiveness.

**Strengths:**

* The paper is clearly written and easy to follow.
* The proposed approach is intuitive and readily adaptable across different types of VLLMs.
* Exploring joint frame-level and token-level compression represents a promising direction for efficient video understanding in VLLMs.

**Weaknesses:**

* While the idea is intuitive, it is also relatively straightforward, as both frame-level keyframe selection and token-level visual token pruning have been explored in prior work. Given this, a more comprehensive ablation study would be essential to provide deeper insights. Unfortunately, the current experimental setup offers limited analytical value in this regard.
* The performance gains reported are not consistently substantial. In certain configurations, such as LLaVA-Video on VideoMME with subtitles, the improvement is marginal and may not constitute a statistically or practically meaningful gain.
* Lack of comparison with the training-free methods developed for image LLMs, such as IG-VLM [1], SF-LLaVA [2], TS-LLaVA [3]. This type of methods usually only uses token compressions with uniformly sampled frames. And experiments on more datasets, e.g. MLVU [4], LongVideoBench [5], TempCompass [6], EgoSchema [7] etc., are also expected to draw a clear conclusion.

[1] Wonkyun Kim, Changin Choi, Wonseok Lee, Wonjong Rhee. (2024) An Image Grid Can Be Worth a Video: Zero-shot Video Question Answering Using a VLM.

[2] Mingze Xu, Mingfei Gao, Zhe Gan, Hong-You Chen, Zhengfeng Lai, Haiming Gang, Kai Kang, Afshin Dehghan (2024) SlowFast-LLaVA: A Strong Training-Free Baseline for Video Large Language Models.

[3] Tingyu Qu, Mingxiao Li, Tinne Tuytelaars, Marie-Francine Moens (2024) TS-LLaVA: Constructing Visual Tokens through Thumbnail-and-Sampling for Training-Free Video Large Language Models.

[4] Junjie Zhou, Yan Shu, Bo Zhao, Boya Wu, Zhengyang Liang, Shitao Xiao, Minghao Qin, Xi Yang, Yongping Xiong, Bo Zhang, Tiejun Huang, Zheng Liu (2024) MLVU: Benchmarking Multi-task Long Video Understanding

[5] Haoning Wu, Dongxu Li, Bei Chen, Junnan Li (2024) LongVideoBench: A Benchmark for Long-context Interleaved Video-Language Understanding

[6] Yuanxin Liu, Shicheng Li, Yi Liu, Yuxiang Wang, Shuhuai Ren, Lei Li, Sishuo Chen, Xu Sun, Lu Hou (2024) TempCompass: Do Video LLMs Really Understand Videos?

[7] Karttikeya Mangalam, Raiymbek Akshulakov, Jitendra Malik (2023) EgoSchema: A Diagnostic Benchmark for Very Long-form Video Language Understanding

**Questions:**

Please refer to the weaknesses.

---

### Official Review · Reviewer_YEaF · 2025-11-02

**Soundness:** 2
**Presentation:** 2
**Contribution:** 1
**Rating:** 2
**Confidence:** 4

**Summary:**

The paper proposes a compression scheme for the visual tokens in a VLM, to reduce computational complexity while maintaining (or even slightly increasing) performance. The proposed scheme consists of 2 steps: first frames are selected ('pixel-level'), using a simple L1-norm based comparison of frames. Then tokens with high mean similarity to other tokens are pruned ('token-level'), with a dynamic pruning ratio.
Results are reported for both a training-free as well as a finetuning setup, using the MVBench benchmark.

**Strengths:**

1. The proposed method is relatively simple and easy to implement. This also makes it easy to integrate with various VLM ('plug and play').
2. Both training-free and finetuning settings are tested.

**Weaknesses:**

1. The authors are not the first to show the number of tokens for a VLM can be reduced significantly without significant impact on the accuracy.
Focusing on the training-free setting, compared to e.g. DyCoke, they improve somewhat, but it's hard to tell whether this is significant. It could also be due to a somewhat better choice of hyperparameters (by luck, or by trying out a few things as shown in the ablation study and picking the best). Even if it is, the take-home message of the paper is, at best, 'with some tweaking of the token selection, we can improve the process a bit".

2. While the proposed method is relatively simple, the description of the token-level compression seems incomplete and therefore hard to reproduce. For instance,
- it's unclear whether the similarity between tokens is computed before or after positional embeddings are added.
- if the token compression works with a sliding window to partition frames into groups (l. 235), every frame serves as anchor frame at some point ? Or do you work with non-overlapping windows ? Then the term 'sliding window' may be confusing.
- it's unclear how the min/max pruning ratio is enforced, when the threshold tau is predefined (and fixed?).
- I've no idea how to interpret the notation in eq. 4
- an algorithm for the token compression, as provided for the pixel compression, might help clarifying the process.

3. Hyperparameters such as group size, min. and max. pruning ratio, similarity threshold, etc. are not specified. It's also not explained how their value has been determined (especially relevant for the training-free setting) .

4. The different experiments are somewhat redundant. It would be more interesting
- to see a comparison against naive baselines such as randomly dropping frames or tokens;
- compare against other similar methods such as Deco (https://arxiv.org/abs/2405.20985) or LlaMa-VID (https://llama-vid.github.io/)
- including other benchmarks with longer videos such as VideoMME, MLVU, LongVideoBench

5. The text is somewhat repetitive, re-iterating on the same points over and over. It could be shortened significantly.

**Questions:**

1. Are positional embeddings added before the similarity between tokens is computed ?
2. How are the values of the hyperparameters of the method determined ?
3. Can you include a comparison against state-of-the-art methods that are not training-free ?
3. Instead of showing on-par accuracies with lower compute/memory budget, it would be more interesting to see that, on longer videos, you can actually improve accuracy by capturing more of the video content than the default setting that samples a too coarse set of frames.

---

### Meta-Review · Area_Chair_S1gv · 2026-01-07

**Summary:**

This paper received 4 reviews. The reviewers (score/confidence) are: `YEaF (2/4), eJST (2/4), fGcp (2/3), BDB3 (2/3)`. Their major concerns:
- Methodology:
  - The token-level compression scheme lacks complete descriptions, leading to poor reproducibility. E.g., ambiguity in positional embedding timing, sliding window mechanism, and pruning ratio enforcement (`YEaF (2/4)`).
  - The core idea of joint pixel-token compression has limited novelty, as both frame selection and token pruning have been extensively explored in prior works, with insufficient differentiation from existing methods (`BDB3 (2/3)`).
  - The method is heuristic with little theoretical analysis to explain why joint compression preserves semantic fidelity (`fGcp (2/3)`).
  - Hyperparameter settings (eg, group size, pruning ratio, similarity threshold) are not specified, and their determination process is unclear (`YEaF (2/4)`).

- Experiments:
  - The experimental comparison is incomplete: no comparison with naive baselines (random frame/token dropping) and state-of-the-art methods (e.g., Deco, LlaMa-VID, IG-VLM) (raised by many reviewers, such as `YEaF (2/4), eJST (2/4)`).
  - The performance gains are marginal and lack statistical significance in some configurations (e.g., LLaVA-Video on VideoMME with subtitles) (`eJST (2/4)`).
  - No experiments on long-video benchmarks (e.g., MLVU, LongVideoBench) to verify the method's scalability (`YEaF (2/4), eJST (2/4)`).
  - Hyperparameter sensitivity analysis is missing, with limited discussion on generalization across different datasets (`fGcp (2/3)`).

- Presentation:
  - The paper has repetitive content and can be significantly shortened (`YEaF (2/4)`).
  - The ablation study is insufficient to provide deep insights into the joint compression mechanism (`eJST (2/4)`).

The authors did not submit a rebuttal. And thus, no reviewers have changed their score so far. Given the paper's condition and the lack of a rebuttal, I recommend **Reject**.

**Reviewer Concerns:**

The authors did not submit the rebuttal, so the concerns are all outstanding.

**Reviewer Scores:**

The authors did not submit the rebuttal, and the concerns are shared. The score will probably stay at 2/2/2/2.

---

### Decision · Program_Chairs · 2026-01-26

Reject